# The Application of Blockchain-Based Life Cycle Assessment on an Industrial Supply Chain

**Xuda Lin** [1,2,*] , **Xing Li** [3], **Sameer Kulkarni** [4] and **Fu Zhao** [1,4]

1   Division of Environmental and Ecological Engineering, Purdue University, West Lafayette, IN 47907, USA; fzhao@purdue.edu
2   Department of Statistics, Purdue University, West Lafayette, IN 47907, USA
3   School of Civil Engineering, Purdue University, West Lafayette, IN 47907, USA; li1856@purdue.edu
4   School of Mechanical Engineering, Purdue University, West Lafayette, IN 47907, USA; kulkar15@purdue.edu
*   Correspondence: lin468@purdue.edu; Tel.: +1-765-(409)-6349

**Abstract:** Life cycle assessment (LCA) is a widely recognized tool used to evaluate the environmental impacts of a product or process, based on the environmental inventory database and bills of material. Data quality is one of the most significant factors affecting the analysis results. However, currently, most datasets in inventory databases are generic, i.e., they may represent the material and energy flow of a process at a market average, instead of a specific process used by a manufacturer. As a result, stockholders are unable to track their supply chain to find out the actual environmental impact from each supplier and to compare the environmental performance of alternative options. In this paper, we developed a new framework, i.e., blockchain-based LCA (BC-LCA), where blockchain technology is adapted to secure and transmit inventory data from upstream suppliers to downstream manufacturers. With BC-LCA, more specific data can be acquired along the supply chain in a real-time manner. Moreover, the availability, accuracy, privacy, and automatic update of inventory data can be improved. A case study is provided based on an industrial supply chain to demonstrate the utilization of BC-LCA.

**Keywords:** life cycle assessment; blockchain; supply chain

## 1. Introduction

In recent decades, environmental problems have been an increasingly global issue, affecting everyone. Typical environmental problems include global warming, chemical pollution, depletion of natural resources, and the loss of biodiversity [1]. The blooming of new technologies and demand of products brings growth of industry, but also leads to more serious environmental problems. Thus, to produce a more accurate and fast quantification of the environmental impacts caused by a certain process, a reliable tool is required, which could measure and calculate the environmental impacts in different aspects [2].

Life cycle assessment (LCA) is a widely used tool to analyze environmental impacts. The users of LCA include governments, non-governmental organizations, industrial sectors, and academic and education institutes. The results of LCA could provide customers with a reference so that they can make comparisons. LCA can also benefit decision making by providing information on several alternative products or processes so that there could be some space for trading off [3].

Blockchain, prominently implemented in cryptocurrency Bitcoin [4], could be a promising and complementary addition to LCA. Besides cryptocurrency, blockchain is widely used in financial services, health and medical services, energy rebalancing, agricultural services, and so on. Previous research shows that blockchain can also be applied to improve the sustainable performance as well as the resilience of the supply chain [5,6]. LCA has a very close relationship with the bill of material for products, which are derived from the

supply chain. Thus, the integration of blockchain and LCA theoretically and practically would lead to a more convenient and reliable database.

Blockchain is a data structure, composed of a chain of blocks interconnected in a unidirectional order via an encrypted Hash function. The classic blockchain framework is shown in Figure 1. Each block contains three parts: a previous hash, a nonce, and transaction data. The previous hash contains information acting as a pointer to the previous block, allowing us to backtrace the information along the blockchain. The second part is a nonce, which formats the hash of the whole block into a string with a certain number of zeros in the very first several digits. The third part is transaction data, which would be stored in a tree [7].

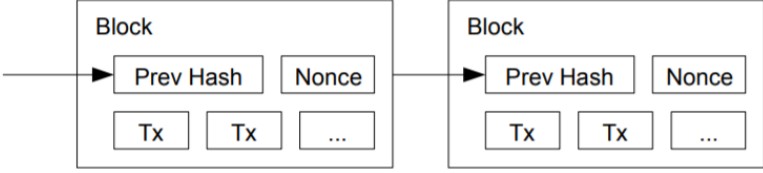

**Figure 1.** Classic blockchain framework in Bitcoin [4].

In the classic blockchain, mining is a process performed to find a nonce such that it can make the hash of the whole block into a string with a certain number of zeros in the very first several digits. Each node must generate a nonce, then perform the SHA-256 (an encipher algorithm), and repeat, until such a nonce is found. This circular process is defined as proof-of-work, as each processor (CPU) can only finish a limited number of SHA calculations. After the nonce is found, the node can pack up the transaction data into a block, using this nonce, and broadcast this block to the whole network, allowing all the nodes in the network to verify if this block is legal (which means this block follows all the rules and restrictions on the blockchain). After the verification, all other nodes would copy the information on that block so that all the transaction data on that block could be applied. The node that finds that nonce and packs up the transaction data would be rewarded with several cryptocurrencies (e.g., Bitcoin or Ethereum).

The research about using blockchain on life cycle assessment is in its early stages. Smetana mentions the impact of the artificial neural network (ANN) and blockchain on revolutionizing material flow analysis (MFAs) and LCAs [8], which is more like a conceptual reference, saying that some new technology, especially ANN and blockchain, could provide a positive improvement in MFAs and LCAs. In 2020, more research about blockchain-related life cycle assessment was published. A framework about the implementation of blockchain-based LCAs was developed [9], with a budget estimation. This study combined blockchain-based LCA with the Internet of Things (IoT), trying to automatically fetch data from sensors but with very limited discussion about the mechanism. A fuzzy DEMATEL analysis on blockchain-based LCA in China was provided [10], which gives a result that blockchain-based LCA could improve manufacture data accuracy. A strategy-related study on blockchain-enabled LCA was published [11], discussing the concern that BC-LCA might obtain support from strategy. In 2021, a conference paper about blockchain-based LCA and its aircraft-related application was published in CIRP (The International Academy of Production Engineering) [12], which provides a good example of BC-LCA application.

This paper provides a much more detailed description of the BC-LCA framework and mechanism, compared with previous research, together with a case study based on a practical industrial supply chain about chemical manufacturing, to quantitively prove the improvement in data availability, data accuracy, and data privacy, so that BC-LCA can better support environment-related strategy and interact with other modern technologies.

This paper is organized as follows. In Section 2, the disadvantages of the current LCA framework that can be fixed by a blockchain-enabled LCA (BC-LCA) would be discussed. Section 3 presents the key assumptions that this study was based on. Section 4 provides the framework and mechanism about BC-LCA, and its benefits would be discussed in

Section 5. A case study discussing the potential implementation of BC-LCA for a chemical generation process is also provided in Section 6.

## 2. Disadvantages of Current LCA

In any product's supply chain, there are many enterprises (suppliers, shipping, manufacturing, raw material acquisition, etc.), which are considered "nodes". The partnership in-between two nodes is an "edge" [13]. LCA often faces challenges in collecting reliable data from the supply chain, especially from a supply chain with multiple nodes and complex edges. Theoretically, the results of LCA only depend on two critical parts: the bills of material and environmental inventory data. Obtaining reliable data is a very challenging problem [14]. In classic LCA, each node should obtain the environmental inventory data on its own. Currently, most facilities would use published environmental inventory databases such as ecoinvent. All the data are provided by researchers under some assumptions (e.g., the scope definition), but when these data are implied, these assumptions might be ignored by the user. In fact, not all the LCA research can find appropriate data. The practical LCA application needs the database that has higher accuracy and higher availability and is more specific to the product or process being studied.

### 2.1. Inefficiency of Data Transmission

The most popular environmental inventory databases are always from centralized companies and organizations (e.g., Sphera, ecoinvent). These databases are collected and compiled by a specific organization first and then published to users all over the world. Theoretically, all the data transmission in a centralized structure should go through a center. This center could be a facility (e.g., a bank in wired transfer process) or a network (e.g., cellular network in the process of a phone call) and should process all the restrictions and the verifications. It might be too abstract to directly talk about the data transmission efficiency without any example. The banking system is one of the most typical centralized structures, and it is close to daily life and easy to explain. If Customer A wants to transfer three dollars to Customer B, the bank needs to check if there is enough money on A's account and whether the account information about B is correct. All the restrictions and verifications happen in the bank, which makes it easy to update the rule. However, wired money transfer may take up to days, and international money transfer may take up to weeks. This is a tradeoff for the efficiency in using centralized structures. On the one hand, considering restriction and verification processes, the centralized structure is efficient enough and powerful, because all the restrictions and verifications are easy to imply [15]; on the other hand, there is a significant disadvantage commonly shared by centralized structures: data transmission inefficiency [16].

The inefficiency of a centralized structure in data transmission is worse in LCA data exchange, as there is no such a shared centralized platform on which to perform the data collection. The procedure of data updating in LCA is much more inefficient: If Researcher A has done some LCA research and updated some data, Researcher A must publish a paper to describe this update. Then, the environmental inventory data companies and organizations (e.g., PRé, ecoinvent, Sphera) must keep reading papers to catch up with this kind of updates, so that the database companies and organizations can then record these updates into a new version of the environmental inventory database. Eventually, all the users could see this update after the new version is released by the database companies and organizations. Figure 2 shows the whole process based on the previous example: if any node in a supply chain wants to perform LCA, it is necessary to download the environmental inventory data from a centralized data company. Meanwhile, the LCA results would be published via papers or other publications, which could be used as a data source for the companies or organizations that publish environmental inventory databases.

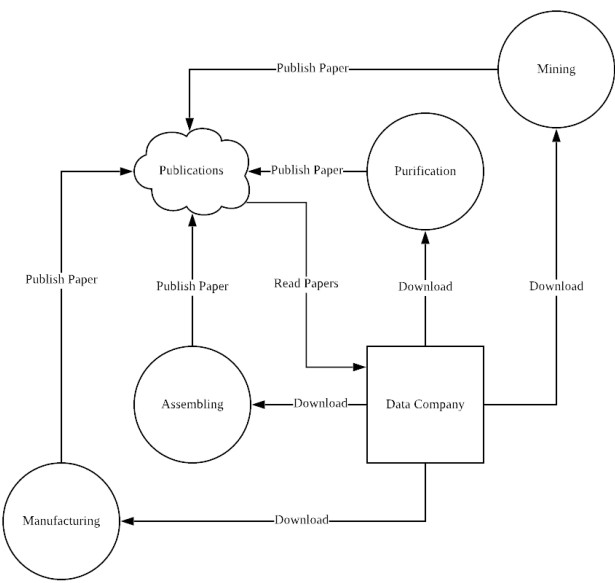

**Figure 2.** Centralized LCA structure.

## 2.2. Lack of Data Availability

The current environmental inventory database contains significant information about different kinds of manufacturing processes. The recent ecoinvent database contains more than 16,000 datasets [17]. For researchers, if generic environmental inventory information is needed about one generic type of process, then the aggregated (average) environmental inventory dataset might be enough. For example, if researchers are interested in the comparison between two different types of products, such as a paper cup and a polymer cup, then the generic data might work. However, if a user wants to know the environmental impact of one specific process, then generic information is not enough. For example, if a user wants to know the environmental performance of a laptop with a specific model, such as HP Envy X360 with 8 GB RAM from Kingston, 1T HDD from Western Digital, or Core i5-8500 processor from Intel, then generic data can only provide the user with the environmental impact of a generic laptop, instead of this specific model. With the current database, a user will never know the accurate environmental impact of one specific product or process.

## 2.3. Concerns about Data Privacy

According to ISO 14044, the results of LCA depend on two critical parts: the environmental inventory database and bills of material. In most cases, environmental inventory data are provided by public organizations or companies, which is a typical centralized structure. The reason behind this is that manufacturers are hesitant to share their data, no matter if it is their inventory data or environmental inventory data of their final products. Even within one company, the data privacy problem also can exist between different departments. The priority of protecting their own data is much higher than sharing these data to benefit the whole supply chain. Thus, probably the only way to solve this problem is finding a trustworthy third party (e.g., a data company) to manage these data and make the data anonymous to all database users. However, there is still the possibility that this third party would leak out the name or bills of material to the public. Therefore, few manufacturers are willing to share their data.

## 3. Key Assumptions

Current circumstances do not only depend on the platform or engineering, but also on the natural unwillingness toward data sharing. Companies or organizations are more concerned with personal gains or losses during data sharing, instead of how data sharing can benefit them and the whole industry. It is understandable that companies or organizations

are responsible for keeping their own data confidential. However, this is one of the most important factors that impedes a good data-sharing atmosphere. To solve this problem, an engineering solution is not sufficient. A more powerful and effective administrative solution should be developed to help guide companies to a beneficial solution.

There are three major assumptions that this study is based on:

1.  Companies or organizations are willing to share their data and improve the environmental performance.
2.  Economic consideration is not the only threshold to consider during the selection of alternatives. Environmental impact also plays a significant role, and the product with better environmental performance could be more attractive.
3.  Stakeholders are willing to pay more for environmentally friendly products.

Under these assumptions, this study provides a solution to performing automatic LCI data transmission and LCA calculation.

## 4. Framework and Mechanism

In this study, by applying blockchain technology, BC-LCA can replace the traditional centralized structure and improve the efficiency of data transmission. In BC-LCA, all the data verifications and restrictions can be performed by an arbitrary node, instead of a fixed "center" (Figure 3), thereby drastically improving the efficiency of data transmission. However, there is one thing that needs to be considered: every node should contain the rule to perform restrictions and verifications. In other words, if a manufacturer decides to join the BC-LCA, this manufacturer should act as a node with the full function to perform the verifications and imply all the restrictions. Thus, BC-LCA requires a more meticulous design than the traditional LCA framework. BC-LCA can overcome the limitation of the centralized structure and improve the performance of classic LCA.

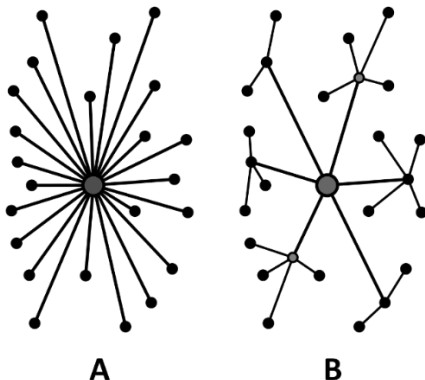

**Figure 3.** Graphical comparison of a centralized (**A**) and a decentralized (**B**) system.

Based on the classic LCA framework and classic blockchain framework (Figure 1), the new blockchain-based LCA network could have the full function in both LCA and blockchain. It could analyze the environmental impact of a product or process, and all the data transmission would go through the blockchain. The new blockchain-based LCA framework would also be more automatic and accurate.

Figure 4 shows the framework of BC-LCA. In every block, "Previous Hash" and "Nonce" act as a linkage between blocks. "Material Flow" and "Environmental Performance" data would be stored in each block. In this case, every time a material flow occurs, this data would be broadcast into the BC-LCA network, and the environmental performance of this material flow would be sent directly to the downstream node (which is the receiver). Each block has its capacity limitation, which means one block can store a limited number of material flow data and environmental performance data. Thus, after one block is filled up, an arbitrary node would appear and "seal" the whole block and broadcast the block to the whole network.

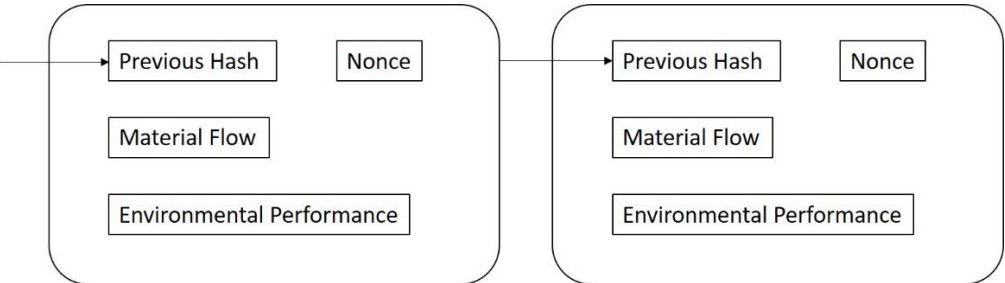

**Figure 4.** Framework of BC-LCA.

The mechanism of BC-LCA is simple but powerful. All information would be transferred via broadcast to the whole BC-LCA network. There are three types of nodes that participate in this information transfer: the sending node, the receiving node, and transfer nodes. The sending node is the one that broadcasts information into the BC-LCA network, as well as the node that sends out material flow. The receiving node is the node that receives the material flow. Transfer nodes are these nodes that only participate in information transfer but are not the sender or the receiver nodes.

Figure 5 shows an example of information transfer. Node Mining wants to transfer 1 ton of Product A to Node Purification, which is defined as "material flow". Simultaneously, Node Mining would broadcast this information to the whole network with the following procedure:

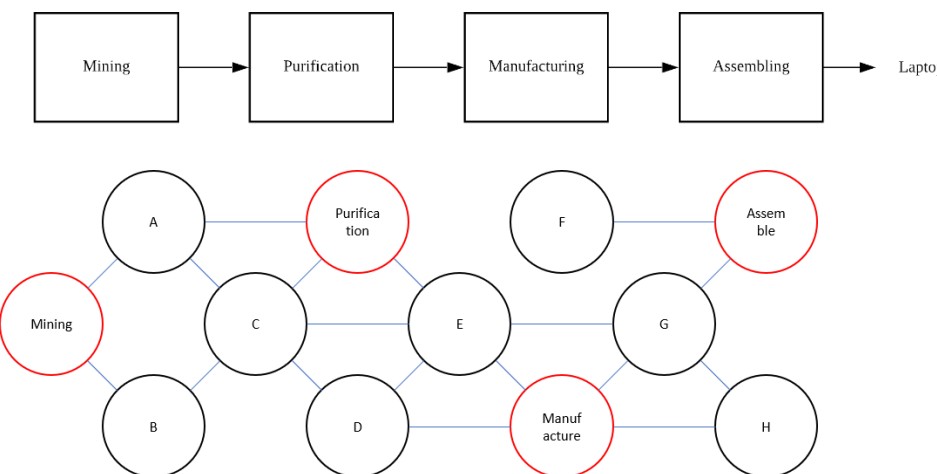

**Figure 5.** Decentralized LCA network sample.

1. Getting information ready: Node Mining will need to prepare the list of products (1 ton of Product A) and environmental impacts related to these products (environmental impacts of Product A).
2. Notify Node A and Node B: The closest nodes of Node Mining are Node A and Node C, which would be the aim of broadcasting. Here, Node Mining would send out the prepared information to Node A and Node B directly.
3. Node A would notify Node Purification, and Node B would notify Node C; however, the closest nodes for Node A are Purification and C and the closest node of Node B is C. Therefore, Node A will send information to Node Purification and Node C, and Node B will also send information to Node C. It does not matter who will send Node C the information first, as finally, both Node C and Node Purification obtain the necessary prepared information.
4. Node Purification receives environmental impacts; all other nodes keep the record. After the prepared information is received by Node Purification, Node A, Node B, and Node C would always keep the prepared information. Furthermore, the prepared

information would keep spreading until it is received in all the nodes in the BC-LCA network. Here, besides Node Mining and Node Purification, all other nodes would keep the prepared information for record and to prevent cheating.

There would be a large amount of information spreading in the BC-LCA network at the same time. However, it is not necessary to worry if one piece of information would be transferred duplicated between one node pair. Every time a node receives a piece of information, the node would check if this information already existed. Any node would only accept the information that is not received. When the last node in BC-LCA receives the information, the node would find no other node to send to; then, the whole broadcasting is over.

In BC-LCA, any node can perform the LCA calculation. At the time a node receives material flow information, it would automatically calculate the environmental impact. Here, the update of products' environmental performance occurs every time this node receives material flow information. The calculation would consist of two parts: bills of material and LCI data. Both parts come from the BC-LCA network and could be auto-updated if the upstream nodes provide update information. This calculation could directly provide the results of the product's final environmental inventory table and broadcast to the whole BC-LCA network.

In addition to the external-node data transition, there are three pre-processes that should be considered in the internal-node calculation:

1. Bills of material confirmation: The very first step before calculation is to make sure the input data are valid. Here, a node needs to check whether the input bills of the material table are legal and whether each element of this table has an appropriate format.
2. LCI dataset confirmation: The second step is to confirm the LCI dataset is up-to-date and contains all the information to calculate. Each node should have multiple versions of the LCI dataset, including the dataset directly from upstream nodes and the dataset for recording only. Before any calculation, a node should search for and sort out the appropriate LCI dataset.
3. Unit consistent inspection: After confirming bills of the material and LCI dataset, the next step should be checking the unit consistency. A different unit could lead to a significant impact on environmental performance factors. The unit in bills of material should match the unit in the LCI dataset.

After these pre-processes, a node could perform a matrix multiplication [18] to figure out the environmental impact results of one product. Multiple products need multiple calculations, with all three pre-process steps for each calculation. When a node finishes an environmental impact analysis for one product, this node can then broadcast the results to the whole BC-LCA network. Then, all downstream nodes, which may be impacted by this analysis, could update their related products. The whole process, starting with the first node that changes its product environmental impact data until the last node finishes updating, is called cascading.

## 5. Benefits of BC-LCA

### 5.1. Data Availability

To make the accurate environmental inventory data available for a specific product, the first step is to acquire the accurate data on every item in this product's bills of material, which means the upstream nodes (i.e., suppliers) should provide accurate information for their products. The easiest way to let the supplier provide the environmental inventory data for their products is by bringing the supplier into the same blockchain network. In this case, while material goes from upstream to downstream, the environmental inventory data should follow the same direction. This is the meaning of a blockchain-based LCA network: digitization of all the material flow and providing accurate data to all the nodes.

*5.2. Data Privacy*

Benefited by the cryptocurrency, blockchain technology has already had a practical enciphering algorithm: the Security Hash Algorithm (SHA), published by the National Institute of Standards and Technology (NIST) as a U.S. Federal Information Processing Standard (FIPS) [19]. This method can promise that enciphered private information can only be read or edited by the user who has the key, with no exception. The key is a string with the information to decipher the user's private data. Even the enciphered data owner cannot read the enciphered data without the key. By doing this, the user's privacy could be protected, and they could be more willing to share their environmental inventory data of their products to the downstream users.

Here, recall Figure 5 as an example. When Node Mining sends 1 ton of Product A to Node Purification, a key, which contains the information to decipher, would be sent from Node Mining to Node Purification. Then, the transition of material happens, and transition information (1 ton of Product A and the environmental impact of 1 ton of Product A) would be broadcasted to the whole network. However, during this step, the product list (1 ton of Product A) is enciphered, but the environmental inventory information (the environmental impact of 1 ton of Product A) is open to the public. The sender and receiver are also anonymous, which means that all other nodes (except Node Mining and Node Purification) can only see that Node X sends Node Y some unknown product, together with a certain known amount of environmental impact, and Node Purification, which has the key from Node Mining, can decipher the broadcasting information, know that this transmission includes 1 ton of Product A, and calculate the environmental impact of 1 unit of Product A.

*5.3. Cheating Prevention*

In the long term, all nodes should have a balanced input and output environmental impact, even considering the inventory of storage. With BC-LCA, preventing cheating becomes easy: any node can calculate the input and the output environmental impact and therefore can tell whether there any cheating has occurred. Just like the Bitcoin network, a user can know the budget in his/her account, and any BC-LCA user is able to know what the accumulative environmental impact is in its node. Because every transaction in BC-LCA has a timestamp, it would be easy to track the growth/decay rate of accumulative environmental impact and use an algorithm to judge if that node is cheating or not.

## 6. Case Study Based on an Industrial Supply Chain

Willems provides some simplified supply chain data [20], which could be used to demonstrate the application of BC-LCA. Here, an industrial organic chemical supply chain is selected, whose picture is shown in Figure 6.

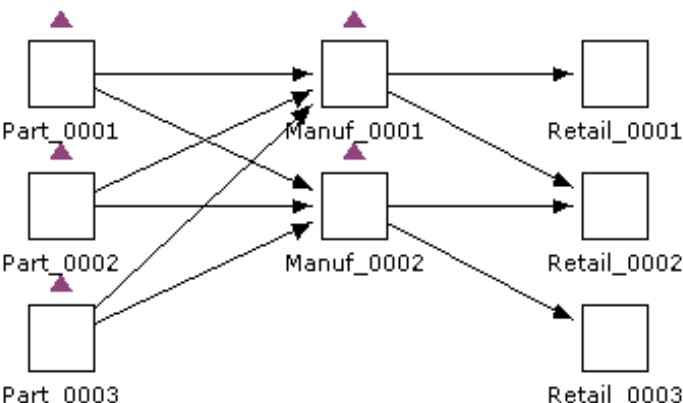

**Figure 6.** Sample supply chain picture [20].

This supply chain includes three procure stages, two manufacture stages, no transportation stage, and three retail stages. In total, there are eight nodes with 10 edges. This

supply chain is comparatively simple, but it could show the difference with and without BC-LCA application. Here, a Python-based BC-LCA is developed for simulation.

Due to the lack of material flow information in the supply chain sample, five assumptions are made:

1. Manuf_0001 and Manuf_0002 share the same manufacturing recipe.
2. Carbon footprint is calculated using US-EEIO based on cost data provided.
3. All nodes are located in California.
4. Only carbon emission is considered.
5. Transactions (material flow) between any two connected nodes happen once a month.
6. The whole model runs from January 2018 until July 2020.

In this case study, five scenarios are considered:

0. No node joins BC-LCA
1. Only Manuf_0002 joins BC-LCA

   a. The percentage of clean energy used in manufacturing is increasing [21]
   b. Due to the procedure upgrade, material waste is decreasing
   c. The energy consumption for supplement (light, A/C, etc.) varies seasonally

2. Part_0001, Part_0002, Part_0003, and Manuf_0002 join BC-LCA

   a. The energy consumption to fetching raw material is decreasing
   b. The percentage of clean energy used in manufacturing is increasing due to Assumption 3
   c. Due to the procedure upgrade, material waste is decreasing
   d. The energy consumption for supplement (light, A/C, etc.) varies seasonally

3. Only Manuf_0002 joins BC-LCA

   a. The energy consumption is increasing due to the aging of equipment
   b. The energy consumption for supplement (light, A/C, etc.) varies seasonally

4. Part_0001, Part_0002, Part_0003, and Manuf_0002 join BC-LCA

   a. The energy consumption to fetching raw material is decreasing
   b. The energy consumption is increasing due to the aging of equipment
   c. The energy consumption for supplement (light, A/C, etc.) varies seasonally

Because Manuf_0001 never joined BC-LCA, instead, Manuf_0001 always uses classic LCA, and the results stay constant over years. The result of Manuf_0001 could be used as a baseline to compare with Manuf_0002, which contains more conditions, and updated monthly. Based on these scenarios, results are shown as below:

Figure 7 shows the carbon emission of the final product under Scenario 0, which considered no node joining the BC-LCA. All carbon emission data are calculated from US-EEIO v2.0 [22], where a factor is provided so that the stage cost of a node can be directly converted to carbon footprint. Scenario 0 is a baseline where carbon emission would not change, and all results depend on the manufacturing recipe.

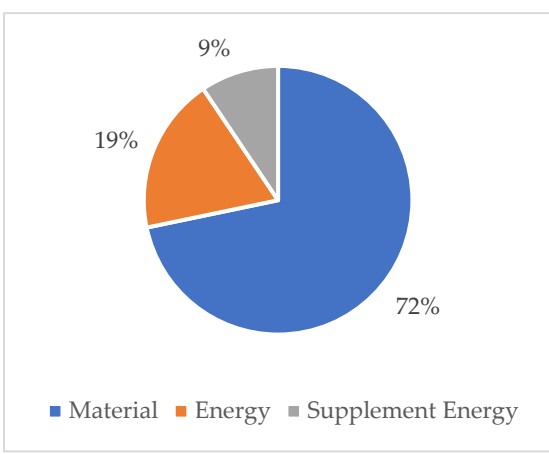

**Figure 7.** Carbon emission of final product under Scenario 0.

Figure 8 shows the carbon emission under Scenario 1. It is obvious that, due to Condition 1a and Condition 1b, the overall trend of carbon emissions is decreasing. Some variance in this curve is due to Condition 1c, which means in every winter, Manuf_0002 would need more energy to maintain room temperature. Compared with Manuf_0001, Manuf_0002 shows accurate seasonal variance.

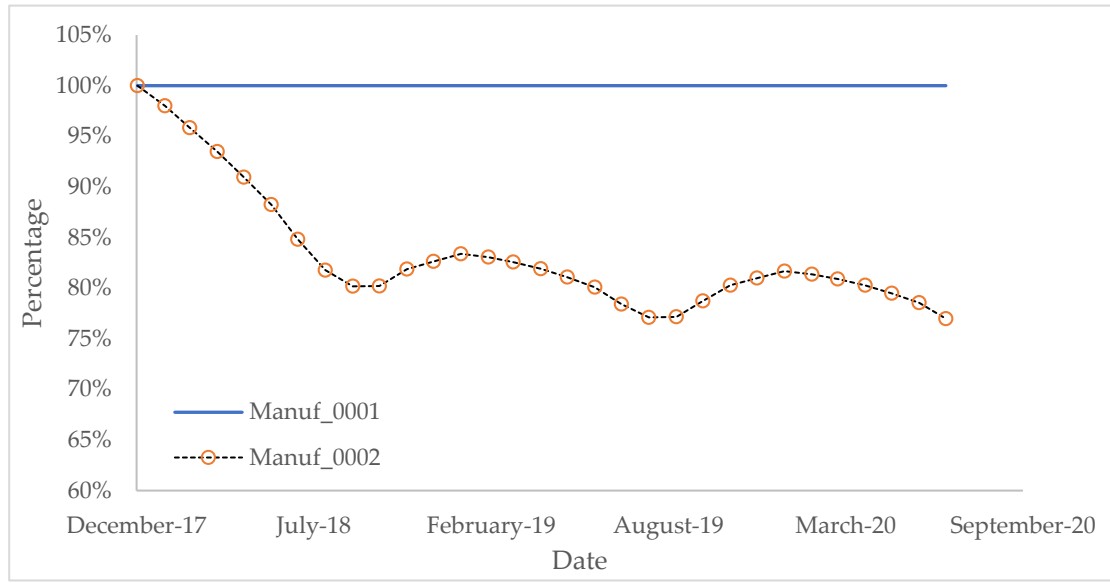

**Figure 8.** Carbon emission comparing percentage of final product under Scenario 1.

For Scenario 2, BC-LCA performs more accurate and variable results. In Figure 9, when more nodes join in BC-LCA, the overall trend of decarbonization is more obvious, which depends on Condition 2a.

With the comparison of Figures 8 and 9, which represent the carbon emission variance when only one node joins BC-LCA and when four nodes join BC-LCA, it would be safe to say that with more nodes in one supply chain joining BC-LCA, the results of environmental impact would be more and more accurate. The reaction to periodical waving and the long-term trend is shown in both Scenario 1 and Scenario 2.

However, the decrease in overall carbon emission is due to the scenario assumptions. The results from BC-LCA could only be more accurate, but not necessarily more environmentally friendly, than traditional LCA. Figure 10 shows the carbon emission results under Scenario 3, where the overall trend of increasing carbon emissions is mostly due to Condition 3a.

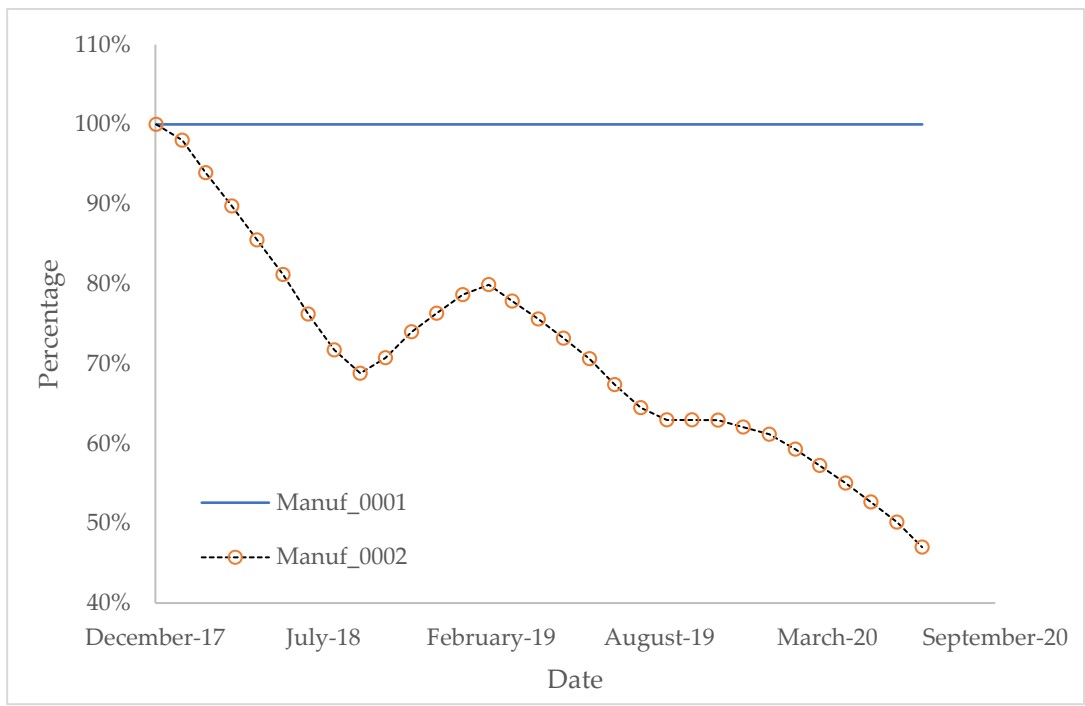

**Figure 9.** Carbon emission comparing percentage of final product under Scenario 2.

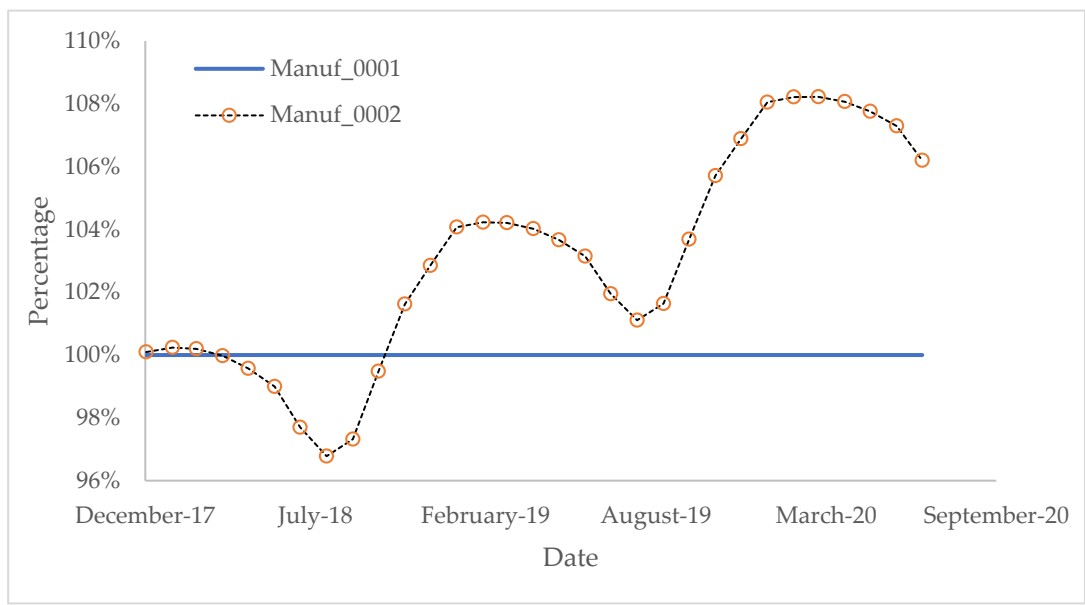

**Figure 10.** Carbon emission comparing percentage of final product under Scenario 3.

Figure 11 shows that, under Scenario 4, even though the energy consumption in Manuf_0002 is increasing due to the aging of equipment, the overall carbon emission is still decreasing.

Based on these scenarios, some comparisons are made (Figure 12). The analysis results show that even this case study implies a simple supply chain, and the difference between BC-LCA and classic LCA is significant. Carbon emission results are more accurate and specific. Manuf_0001 carbon emission could be seen as an average, which is good for general analysis. When Manuf_0002 joins the BC-LCA (Scenario 1 and 3), it is obvious to see how energy consumption, material waste, and supplement energy consumption affect carbon emission. When Part_0001, Part_0002, and Part_0003 also joined in BC-LCA

(Scenario 2 and 4), which means the whole supply chain joined BC-LCA, more factors are considered, so more variances could be observed.

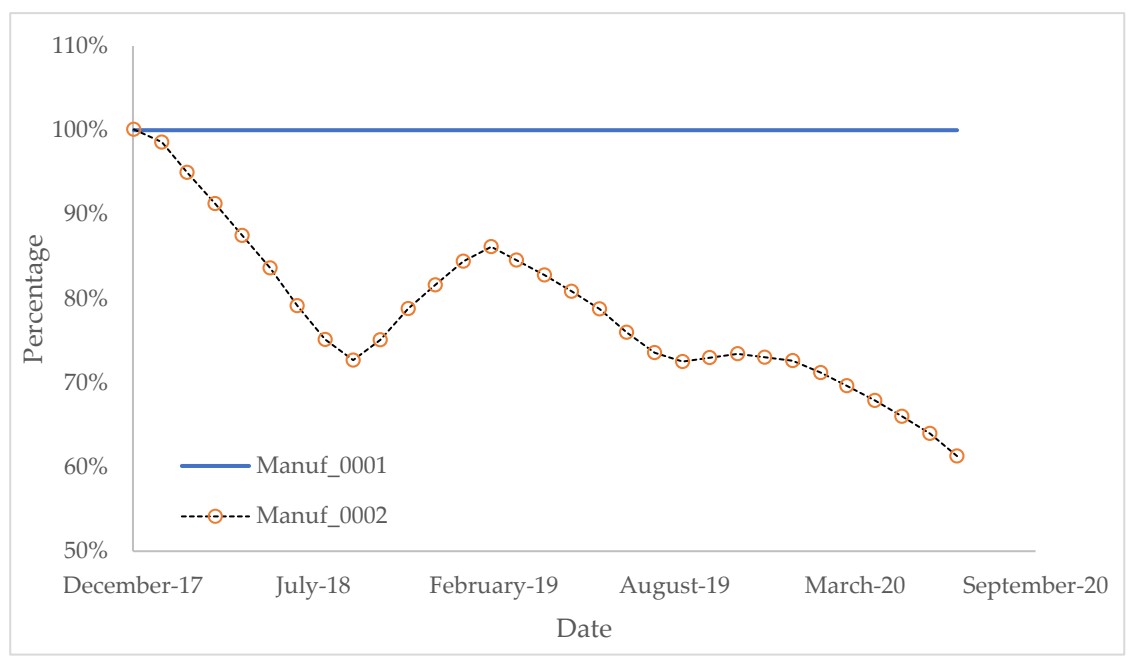

**Figure 11.** Carbon emission comparing percentage of final product under Scenario 4.

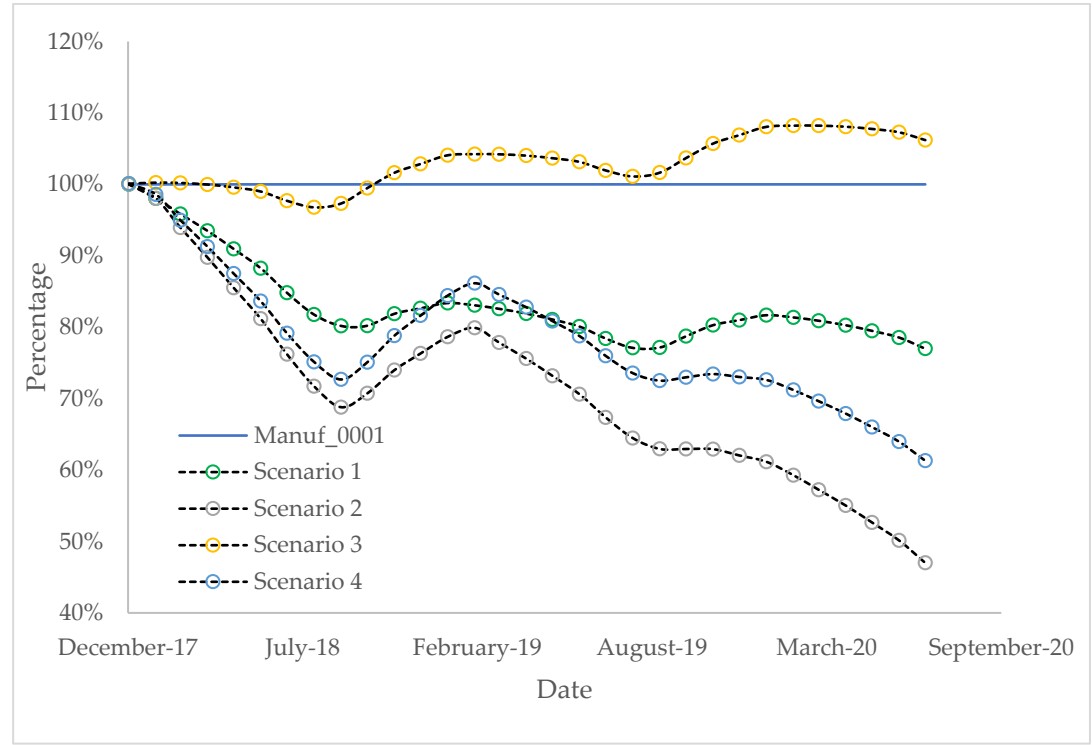

**Figure 12.** Carbon emission comparison of all scenarios.

During the whole case study, all data transmissions are under the blockchain rule. As mentioned before, bills of materials would be enciphered, but total environmental impact of one material flow would be open to the public. The strength of SHA-256 is much higher than a classic security method such as username and password [23].

Tables 1 and 2 show the result of enciphering, which is manually exported and formatted from BC-LCA, based on Scenario 4. Generally, when the BC-LCA network is running, the result of enciphering would be staying in the catcher and would not be that easy to observe. The encipher and decipher operation would be run automatically. Comparing Tables 1 and 2, the name of sender and receiver is constant, though anonymous. The material name after enciphering is dynamic, which means every transaction has its unique key to decipher to enhance data privacy.

**Table 1.** Sample transmission data without encipher.

| Date | Sender | Receiver | Quantity | Unit | Material | Carbon |
|------|--------|----------|----------|------|----------|--------|
| January-18 | Part_0002 | Manuf_0002 | 12 | Dollar | Material 1 | 76.31688 |
| Feburary-18 | Part_0002 | Manuf_0002 | 12 | Dollar | Material 1 | 76.31688 |
| March-18 | Part_0002 | Manuf_0002 | 12 | Dollar | Material 1 | 76.31688 |
| April-18 | Part_0002 | Manuf_0002 | 12 | Dollar | Material 1 | 76.31688 |
| May-18 | Part_0002 | Manuf_0002 | 12 | Dollar | Material 1 | 76.31688 |
| June-18 | Part_0002 | Manuf_0002 | 12 | Dollar | Material 1 | 76.31688 |
| July-18 | Part_0002 | Manuf_0002 | 12 | Dollar | Material 1 | 76.31688 |

**Table 2.** Sample transmission data with encipher.

| Date | Sender | Receiver | Quantity | Unit | Material | Carbon |
|------|--------|----------|----------|------|----------|--------|
| January-18 | Bgrws2dNbee6q6aIaOsf | FzYNXJMtz0UbrAyDvrij | O4UKAYzZuA | Dollar | YpMJUyhQEq | 76.31688 |
| Feburary-18 | Bgrws2dNbee6q6aIaOsf | FzYNXJMtz0UbrAyDvrij | TOx3cUVgxl | Dollar | zVjKPSjWVB | 76.31688 |
| March-18 | Bgrws2dNbee6q6aIaOsf | FzYNXJMtz0UbrAyDvrij | rFxnZB7ucf | Dollar | osIg5wtyXP | 76.31688 |
| April-18 | Bgrws2dNbee6q6aIaOsf | FzYNXJMtz0UbrAyDvrij | tMSkZ1BL7x | Dollar | 2qWqlOpiiJ | 76.31688 |
| May-18 | Bgrws2dNbee6q6aIaOsf | FzYNXJMtz0UbrAyDvrij | n7XXrxdorE | Dollar | QWWMvygVbs | 76.31688 |
| June-18 | Bgrws2dNbee6q6aIaOsf | FzYNXJMtz0UbrAyDvrij | 5VXDudTF7c | Dollar | lu0KP0eM6H | 76.31688 |
| July-18 | Bgrws2dNbee6q6aIaOsf | FzYNXJMtz0UbrAyDvrij | qzNTN8oqs8 | Dollar | Ev1G09lPC2 | 76.31688 |

Another test is also performed to examine the ability of cheating prevention, based on Scenario 4. Manuf_0002 is manually set to cheat, and it claims that its products have a 10% lower carbon footprint than they actually have. In this case, some carbon footprint would be accumulated in Manuf_0002. Additionally, a threshold is set to make sure that any node with higher accumulation carbon footprint than the threshold would be warned and kicked out.

Figure 13 shows the carbon accumulation percentage compared to the threshold. The carbon footprint is always accumulated, and when it reaches the threshold in July 2019, Manuf_0002 is warned and kicked out of the BC-LCA network. The threshold could be modified according to actual requirements.

The whole case study is programmed with Python, where a simple version of BC-LCA is developed, with limited nodes and transactions. However, this framework could be expanded to a supply chain across multiple companies or multiple departments within one big company. The implication of BC-LCA across companies/departments would need every company/department to act as a node, contributing to data broadcasting, transaction ledging, and cheating prevention. Each company/department can save some budget on LCA, as it would be unnecessary to purchase LCA software and database. However, hardware and operators are needed, and BC-LCA may require a more powerful processor due to enormous computational complexity, compared with traditional LCA software.

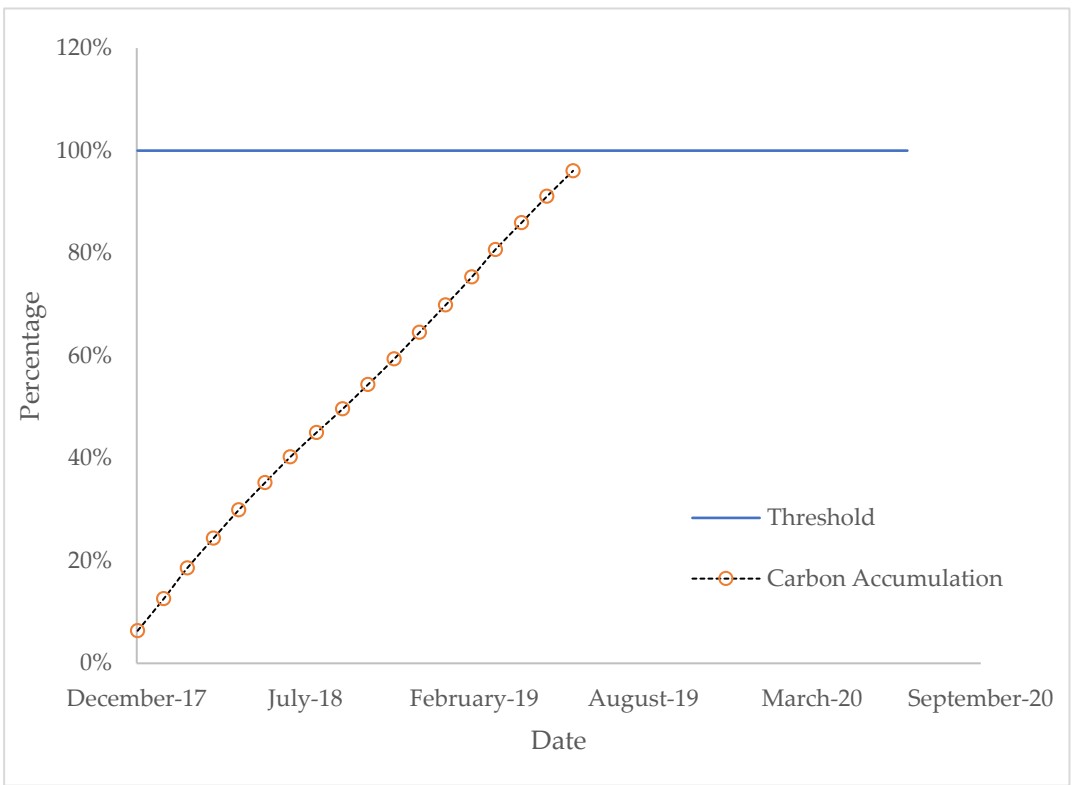

**Figure 13.** Carbon accumulation percentage when cheating.

### 7. Conclusions and Future Work

BC-LCA is the combination of blockchain technology and life cycle assessment, which could significantly improve the availability, privacy, accuracy, and timeliness of LCI data, with less manual operation and time cost. The framework and mechanism of BC-LCA are both designed based on blockchain and modified according to life cycle assessment features. The whole network can automatically calculate environmental impact, transfer data, back up data, and prevent cheating.

One of the properties in BC-LCA is interesting: with more nodes, which belong to the same supply chain, joining in BC-LCA network, data availability and data accuracy would increase. This indicates a trend in BC-LCA development and expansion. The early stage would be tough and slow, as BC-LCA is not able to provide a dramatic improvement when only one or two nodes join in. However, along with more nodes joining the BC-LCA network, there would be a significant breakthrough in data availability and data accuracy, because data companies/organizations (e.g., ecoinvent, GaBi) would not be the only data source, and more specific data would replace generic data. Theoretically, these two points guarantee the attraction that BC-LCA would have.

The implication of BC-LCA has variate potential. Manufacturing would benefit significantly from BC-LCA, due to its strong dependence on the supply chain. Transportation would also obtain a more accurate environmental impact estimation, as tracking odometer and gas filling history is comparatively easier than back-tracking the supply chain. Additionally, building construction, tourism, and agriculture can improve their environmental performance evaluation with the help of BC-LCA, because these fields have a very close relationship with the supply chain as well.

The next step of this research may include an algorithm and data structure design of BC-LCA, as well as a practical demonstration to test the time and hardware cost when BC-LCA is implemented.

**Author Contributions:** Conceptualization, X.L. (Xuda Lin) and X.L. (Xing Li); formal analysis, X.L. (Xuda Lin); methodology, X.L. (Xuda Lin) and X.L. (Xing Li); project administration, F.Z.; software, X.L. (Xuda Lin); validation, F.Z.; writing—original draft, X.L. (Xuda Lin); writing—review and editing, X.L. (Xuda Lin), X.L. (Xing Li), S.K. and F.Z. All authors have read and agreed to the published version of the manuscript.

**Funding:** This research received no external funding.

**Institutional Review Board Statement:** Not applicable.

**Informed Consent Statement:** Not applicable.

**Data Availability Statement:** Not applicable.

**Conflicts of Interest:** The authors declare no conflict of interest.

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
