# Peer review of "The Application of Blockchain-Based Life Cycle Assessment on an Industrial Supply Chain"

_sustainability, doi:10.3390/su132313332_

Round 1

Reviewer 1 Report

Overall, the authors have a good manuscript. However, in its current form, certain limitations weaken the value of the paper in making a stronger contribution. I discuss my concerns below for the authors to consider:

1) The authors need to add more existing literature and discuss your findings with those relevant articles.

2) The discussion and implication section must more clearly be presented in this manuscript, such as theoretical and managerial implications.

Author Response

Hi, thank you for your review! Here's my reply:

  1. The literature review part has been modified, adding the relationship between the current study and previous papers
  2. The discussion part has been enhanced.

Thank you for your time!

Reviewer 2 Report

I congratulate the authors of this document for the novelty of the phenomenon under study. However, I have some suggestions. for. improving research.

1. In the literature review section it is important to add and connect the topic with the theory of sustainability and sustainable supply chain. Only the conceptualization of the variables is appreciated, but it requires a theoretical foundation through a conceptual model that supports it.

2. In the method section it is important to clarify more precisely the people who participated in the study, whether they were the directors, managers or all employees of the organization. As well as the steps involved in using the methodology for a case study.

3. The conclusions are weak (compare the results with other empirical studies and with the theory that supports the study. In addition, it is necessary to give more explanation of the theoretical and practical implications of the study involving all interest groups. theoretical and empirical contributions of this study?, need to be clarified Add the limitations of the study.

Author Response

Hi, thank you for your review. Here is my reply:

  1. The literature review part is modified, adding the relationship between the current study and previous papers
  2. The contribution from different authors are listed in the Contribution part
  3. The conclusion part has been enhanced.

Thank you!

Round 2

Reviewer 2 Report

With this version of this manuscript, I consider that my previous comments have been solved.